# Sustainable Nanomagnetism: Investigating the Influence of Green Synthesis and pH on Iron Oxide Nanoparticles for Enhanced Biomedical Applications

**DOI:** 10.3390/polym15183850

**Published:** 2023-09-21

**Authors:** Johar Amin Ahmed Abdullah, Álvaro Díaz-García, Jia Yan Law, Alberto Romero, Victorino Franco, Antonio Guerrero

**Affiliations:** 1Departamento de Ingeniería Química, Escuela Politécnica Superior, Universidad de Sevilla, 41011 Sevilla, Spain; aguerrero@us.es; 2Departamento de Física de Materia Condensada, ICMS-CSIC, Universidad de Sevilla, 41012 Sevilla, Spain; adiaz18@us.es (Á.D.-G.); jylaw@us.es (J.Y.L.); vfranco@us.es (V.F.); 3Departamento de Ingeniería Química, Facultad de Química, Universidad de Sevilla, 41012 Sevilla, Spain; alromero@us.es

**Keywords:** nanomagnetic iron oxide particles, membranes, green synthesis, stability, magnetic properties

## Abstract

This study comprehensively analyzed green nanomagnetic iron oxide particles (GNMIOPs) synthesized using a green method, investigating their size, shape, crystallinity, aggregation, phase portions, stability, and magnetism. The influence of pH and washing solvents on the magnetic properties of the nanoparticles and their incorporation into PCL membranes was examined for biomedical applications. Polyphenols were utilized at different pH values (1.2, 7.5, and 12.5), with washing being performed using either ethanol or water. Characterization techniques, including XRD, SEM, TEM, FTIR, and VSM, were employed, along with evaluations of stability, magnetic properties, and antioxidant activity. The findings indicate that both pH levels and the washing process exert a substantial influence on several properties of NMIOPs. The particle sizes ranged from 6.6 to 23.5 nm, with the smallest size being observed for GNMIOPs prepared at pH 12.5. Higher pH values led to increased crystallinity, cubic Fe_3_O_4_ fractions, and reduced crystalline anisotropy. SEM and TEM analyses showed pH-dependent morphological variations, with increased aggregation being observed at lower pH values. GNMIOPs displayed exceptional magnetic behavior, with the highest saturation magnetization being observed in GNMIOPs prepared at pH 7.5 and 12.5 and subsequently washed with ethanol. The zeta potential measurements indicated a stability range for GNMIOPs spanning from −31.8 to −41.6 mV, while GNMIOPs synthesized under high-pH conditions demonstrated noteworthy antioxidant activity. Furthermore, it was explored how pH and washing solvent affected the morphology, roughness, and magnetic properties of GNMIOP-infused nanofiber membranes. SEM showed irregularities and roughness due to GNMIOPs, varying with pH and washing solvent. TEM confirmed better dispersion with ethanol washing. The magnetic response was stronger with ethanol-washed GNMIOPs, highlighting the influence of pH and washing solvent on membrane characteristics.

## 1. Introduction

The increasing interest in nanomaterial manufacturing technology has led to its widespread application in various industrial sectors for diverse purposes [1]. Among these nanomaterials, iron oxide nanoparticles have gained significant attention due to their exceptional thermal, optical, electronic, and superparamagnetic properties [2]. Iron oxide nanoparticles occur naturally and can be synthesized using different methods, resulting in various iron oxides and oxyhydroxides with distinct crystal structures and iron valence states [3,4,5]. Of particular interest are their magnetic properties, which include saturation magnetization (*M_s_*), coercivity (*H_c_*), remanence (*M_r_*), magnetic diameter, magneto-crystalline anisotropy constant, magnetic relaxation mechanism, and blocking temperature [6]. These nanomagnetic iron oxide particles (NMIOPs or Fe_x_O_y_-NPs), composed of magnetite (Fe_3_O_4_), hematite (Fe_2_O_3_), and/or maghemite (γ–Fe_2_O_3_), exhibit superparamagnetic behavior at sizes below 20 nm, displaying no magnetization in the absence of an external magnetic field [7].

Moreover, their high surface-to-volume ratio and nanoscale dimensions grant them enhanced binding capacity and stability in solution. Due to their biocompatibility, non-toxicity, and unique properties, NMIOPs have found extensive applications in electronics, biomedicine, catalysis, environmental remediation, aerospace and defense, construction, healthcare, agriculture, textiles, and food-related fields [8]. Furthermore, incorporating nanoparticles into polymer-based material offers solutions to various challenges due to their exceptional functional properties [9]. Metal–polymer nanocomposites combine the unique optical, catalytic, electrical, and magnetic characteristics of metal nanoparticles with polymer nanofibers, garnering significant attention. Nanoparticle-based metal oxide has been integrated as nanofiller into polymer-based membrane mats to overcome various shortcomings, including but not limited to oxidative and bacterial deterioration [10].

NMIOPs possess magnetic, specific, unique, and biocompatible features. They inhibit the growth of foodborne pathogens [11] and selectively damage bacterial DNA and proteins while sparing non-bacterial cells [7]. NMIOPs are safe and non-cytotoxic and have the potential to be used as an oral therapy for iron deficiency [12]. Traditional production methods using chemical-reducing agents may introduce impurities and hazardous substances, but a greener approach using polyphenol-rich plant extracts produces purer, smaller, non-toxic, stable nanoparticles with enhanced functionality [13]. 

The magnetic properties of NMIOPs can be influenced by various factors, including the synthesis method, coating, and sample preparation for magnetic measurements. Recently, significant efforts have been devoted to the development of environmentally friendly synthesis approaches, known as “green methods”, to improve the quality of nanoparticles in terms of monodispersity, crystalline structure, tunable size, and high purity [14]. These eco-friendly methods employ plant or fruit extracts, and bio-organisms, utilizing the abundance of phenolic compounds present in these natural sources as reducing agents for metal salts [15]. This approach offers a cost-effective, efficient, rapid, and non-toxic route to produce NMIOPs with diverse dimensions, morphologies, and magnetic properties. However, to ensure their compatibility with living systems and easy removal, preventing agglomeration by reducing particle size is crucial [16,17]. Notably, the magnetic behavior of these nanoparticles can be modulated from superparamagnetic to ferromagnetic by controlling their size. Various parameters, including the phenolic nature of the plant extract, extraction method, reactant concentrations, metal salt, pH, temperature, reaction time, washing process, and calcination temperature, can be tailored to regulate the properties of the resultant nanoparticles [18]. Despite the growing interest in magnetic nanoparticles, many publications lack comprehensive magnetic characterization, hindering comparisons and evaluations. Therefore, this study aims to provide detailed procedures for determining the magnetic properties of nanomagnetic iron oxide particles, which are essential for various applications, including biomedical, electronic, and food-related applications.

This study utilized *Phoenix dactylifera* L. as a rich source of polyphenols to synthesize magnetic iron oxide nanoparticles (GNMIOPs). The investigation focused on evaluating the impact of pH variation and the choice of washing solvent during the synthesis reaction on the structural characteristics, aggregation behavior, morphology, size, phase composition, and magnetic properties of the nanoparticles. Polyphenol-reducing agents were used at three distinct pH values (1.2, 7.5, and 12.5), followed by ethanol washing. The sample with the highest magnetic response (pH value of 7.5) was chosen as the reference and further washed with H_2_O. We carried out a thorough characterization process utilizing a range of techniques, including X-ray diffraction (XRD), Fourier transform infrared spectroscopy (FTIR), transmission electron microscopy (TEM), scanning electron microscopy (SEM), and vibrating-sample magnetometry (VSM). Additionally, we evaluated the zeta potential, magnetic characteristics, and antioxidant activity (DPPH) of the GNMIOPs. Two selected samples (GNMIOPs prepared at pH 7.5 and washed with ethanol or water), based on their magnetism and aggregation behavior, were incorporated into PCL membranes and further characterized in terms of their distribution, morphology, roughness, and magnetism. The magnetic properties, stability, and antioxidant activity of GNMIOPs hold significant importance for a wide range of valuable applications.

## 2. Materials and Methods

### 2.1. Materials

The primary reagents employed in this study were 98% iron (III) chloride hexahydrate (FeCl_3_·6H_2_O), methanol (CH_3_OH), 99.9% ethanol, NaOH, gallic acid (C_7_H_6_O_5_), 2,2-diphenyl-1-picrylhydrazyl (DPPH), 99.9% dimethyl sulfoxide anhydrous (DMSO) (C_2_H_6_SO). Chloroform (Friendemann Schmidt, Parkwood, WesternAustralia) and N,N-Dimethylformamide (DMF) (CH_3_)_2_NC(O)H (Merck, Darmstadt, Germany) were also employed, and Polycaprolactone (PCL) with a molecular weight (Mn) of 80,000 (purchased from Sigma Aldrich, Saint Louis, MI, USA) was the main polymer used. All other chemicals and reagents utilized were of analytical grade.

We sourced *Phoenix dactylifera* L. from palm trees in Seville, Spain, where the climate typically ranges from 6 °C to 36 °C with average temperature of 19 ± 4 °C and average relative humidity of 53% (data from the National Oceanic and Atmospheric Administration). Then, it was left to dry for 44 days in the shade, where the mean temperature was 22 ± 3 °C, and the average RH was 35%.

### 2.2. Nanoparticle Synthesis

We synthesized GNMIOPs following our previous protocols with some modifications [19]. Briefly, the synthesis was carried out by adding 20 mL of reducing agent dropwise into 20 mL of 1 M iron chloride solution (FeCl_3_·6H_2_O). The pH of the reaction was adjusted to 1.2, 7.5, and 12.5 by adding either FeCl_3_·6H_2_O or 5 M NaOH solution. The reducing agent used was an extract of *Phoenix dactylifera* L. (green synthesis). The solutions obtained were transferred to beakers and gently heated on a hot plate at 50 °C with continuous stirring for a duration of 2 h. Subsequently, the mixture underwent filtration using Whatman No.1 filter paper and underwent thorough washing with ethanol, a process repeated at least three times.

Furthermore, a separate mixture, prepared at pH = 7.5, underwent washing with distilled H_2_O to assess the influence of the washing solvent. This washing procedure was implemented to ensure the removal of any impurities and foreign particles suspended in the mixture. Beforehand, pretreatment was performed by placing the samples in an oven at 100 °C for 8 h. The final step involved subjecting the samples to a heat treatment at 500 °C for 5 h, aimed at eliminating excess material from the sample, thereby leaving only the nanoparticles for subsequent characterization.

### 2.3. Nanoparticle Characterization

Given the nanometric and crystalline properties of the synthesized GNMIOPs, it was necessary to employ a range of experimental techniques [19,20]. X-ray diffraction (XRD), using a Bruker D8 Advance A25 diffractometer (Cu anode, manufactured by Bruker Corporation and sourced from Madrid, Spain), confirmed the crystalline phase, crystal systems, and size of the GNMIOPs within the 2θ range of 15–70°. The crystallinity degree, which represents the extent to which a material’s structure is organized in a crystalline (ordered) form, was calculated as the ratio of the area under the crystalline peaks in the XRD pattern to the total area under all peaks in the pattern. Mathematically, it can be expressed as Crystallinity percentage or Crystallinity degree (%) = (Area under crystalline peaks/Total area under all peaks) × 100 [21].

The samples were exposed to X-ray at 40 kV beam power and 30 mA current. The X-ray made contact with the samples at a pitch angle of 0.015° and a passing time of 0.1 s. Additionally, the samples were rotating at a speed of 30 rpm. Valuable insights into the morphology and size of the nanoparticles were obtained with scanning electron microscopy (SEM) using a Zeiss EVO scanning electron microscope (Pleasanton, CA, USA) operating at 10 kV. To facilitate the examination, a small amount of powder from each sample was meticulously deposited onto an SEM stub using a conductive adhesive. Afterward, a 10 nm layer of platinum was meticulously applied to the samples to enhance their conductivity. The SEM images were further analyzed using ImageJ software (1.53q; National Institutes of Health, Bethesda, MD, USA). Transmission electron microscopy (TEM) analysis utilizing a Talos S200 microscope (FEI, Hillsboro, OR, USA) at 200 kV facilitated the determination of nanoparticle morphology and size. To capture an adequate quantity of the sample, a carbon-coated copper grid was immersed in the GNMIOP powder. Subsequently, the grid, with the sample, was directly placed in the TEM sample holder for examination under the electron beam. High-resolution images of the nanoparticles were acquired and analyzed using the aforementioned Talos S200 microscope. To obtain crucial structural information, Fourier transform infrared (FTIR) spectroscopy was utilized within the range of 4000 to 400 cm^−1^. The primary focus was on identifying Fe-O bonds, particularly within the fingerprint region of 800–400 cm^−1^. The analysis was carried out using a Nicolet iS50 FTIR spectrometer manufactured by ThermoFisher Scientific in Madison, WI, USA. The spectroscopic measurements had resolution of 0.482 cm^−1^, enabling a detailed investigation of the bonding characteristics present in the GNMIP structure. We conducted hysteresis loop measurements on the samples using vibrating-sample magnetometry (VSM) at room temperature, employing a Lake Shore VSM Model 7407 manufactured by Lake Shore Cryotronics, Inc. (Westerville, OH, USA). Saturation hysteresis loops were recorded under a maximum magnetic field of 1.5 T. To prevent particle rotation during the VSM measurements, we compacted the powdered samples into 5 mm disc shapes (~0.4 mm in thickness) using Ag capsules. Saturation magnetization (*M_s_*), coercivity (*H_c_*), and remanence (*M_r_*) were evaluated. For the evaluation of the zeta potential of GNMIOPs, the particles were initially dispersed in distilled water at a concentration of 4 mg/mL. The dispersion was then subjected to 15 min of sonication to ensure proper mixing. Around 900 µL of the resulting dispersion was subsequently injected into DTS1070 cells for zeta potential measurements. The measurements were carried out using the Malvern Zetasizer Nano ZSP (United Kingdom), with the temperature being maintained at 25 °C. The acquired data were analyzed using Zetasizer Software v. 8.02 and further processed and labeled using OriginLab Pro version 2019b (9.65) for subsequent analysis. 

### 2.4. Antioxidant Potential

To assess the antioxidant activity of the GNMIOPs, the DPPH test was employed, measuring the inhibition of the DPPH free radical. The *IC_50_* value (mg/mL), indicating the concentration required for a 50% inhibition rate, was used to quantify the antioxidant potential. This value was determined by conducting calculations using GraphPad Prism (version 9.0.0 for windows, San Diego, CA, USA, www.graphpad.com, 17 September 2023).

### 2.5. Preparation and Characterization of PCL/GNMIOP Membranes

Solution electrospinning was employed to fabricate PCL/GNMIOP membranes, following the methods outlined in a previous study with slight modifications [22]. Initially, a solution of PCL at a concentration of 10% weight/volume (*w*/*v*) was prepared by dissolving 1 g of PCL in a 10 mL blend of chloroform and DMF, using a volume ratio of 9:1, at room temperature. This solution served as the foundation for the electrospinning dope. Following this, GNMIOPs (prepared under pH = 7.5 conditions and subjected to either ethanol or water washing) were dispersed into the dope solution using ultrasonication, aiming for a concentration of 1.0 *w*/*w*, and this process was conducted for a duration of 2 h. A laboratory-scale electrospinning apparatus from BioInicia, specifically the Fluidnatek LE-50 setup located in Valencia, Spain, was employed to electrospin 10 mL of the dope solution. The electrospinning parameters were set as follows: 12 kV voltage, 13 cm needle distance from a rotating drum collector (200 rpm), and 0.9 mL/h feed rate. Afterward, the electrospun PCL/GNMIOP nanofibers were air-dried, carefully peeled off, and stored in a desiccator for further characterization. TEM and SEM imaging assessed microstructure, morphology, composition, elemental info, and nanoparticle distribution in electrospun PCL/GNMIOP nanofibers. To profile surface roughness, we utilized a confocal interferometric optical microscope (Sensofar S-NEOX; Sky Tech, Bukit Batok, Singapore) utilizing the amplitude of 8 µm and magnification spanning from 20× to 150×, adhering to ISO 4287 standards [23]. Surface roughness is typically evaluated using two parameters: average roughness (Ra) and quadratic roughness (RMS). Ra represents the arithmetic average of the absolute value of profile height deviations over the mean length, while RMS is the root mean square average of the profile height deviation over the mean length. Additionally, the electrospun PCL/GNMIOP membranes were evaluated for their ability to inhibit free radicals using the DPPH assay. The evaluation included testing GNMIOPs washed with either ethanol or water within the PCL solution against DPPH, while a pure PCL solution served as the negative control. The results of the evaluation were reported as the percentage of free radical inhibition (FRIP, %).

### 2.6. Statistical Analysis

Statistical analysis involved IBM SPSS Statistics 26 software (Released 2019. IBM SPSS Statistics for Windows, IBM Corp, Armonk, NY, USA. Version 26.0) Measurements were expressed as means ± SDs from three replicates. One-way ANOVA assessed significant differences, and Duncan statistical analysis determined significance levels (*p* < 0.05).

## 3. Results

### 3.1. XRD

The X-ray diffractograms of GNMIOPs are shown in Figure 1 as a function of pH values and washing solvent (ethanol or water), identifying their crystallinity, crystallite size, and phase composition (planes in red correspond to magnetite, and planes in black correspond to hematite).

The 2θ (°) and the corresponding crystallographic reflection planes (hkl) of the different nanoparticles are shown in Table 1. 

As can be seen, all diffraction peaks in the GNMIOP diffractograms are attributed to the spinel structure [24].

The pH value and the selection of the washing solvent are seen to influence the ratio of various phases and crystal systems (crystalline structures) of GNMIOPs, as depicted in Figure 2 (data are summarized in Table 2). 

At pH = 1.2, the peaks are attributed to 78.2% of the crystalline cubic structure of hematite and 21.8% of polycrystalline magnetite (JCPDS Nos. 00-900-6316, 00-900-5841, and 00-900-6317) [25,26]. With an increase in pH value to 7.5 (washed with ethanol) by adding 5 M NaOH solution, the peaks changed to represent 15.6% of monoclinic hematite and 84.4% of the crystalline cubic structure of magnetite (JCPDS Nos. 00-722-8110 and 00-210-8027) [27]. In contrast, when the GNMIOPs at pH 7.5 were washed with H_2_O, these peaks represented, once again, 62.4% of trigonal (hexagonal axis) hematite and 37.6% of polycrystalline magnetite (JCPDS Nos. 00-154-6383, 00-153-2800, and 00-900-2319) [25,28,29,30,31,32]. With a further increase in the pH value to 12.5 (washed with ethanol), the peaks corresponded to nearly pure magnetite (3.4% of cubic hematite and 94.6% of polycrystalline magnetite) (JCPDS Nos. 00-900-9768, 00-152-8611; standard iron oxide powder diffraction pattern).

The different crystalline sizes of GNMIOPs were calculated using the Debye–Scherrer equation, as previously mentioned. The sizes and crystallinity degrees of the nanoparticles are reported in Table 3 [21].

When the pH was set to 1.2, the GNMIOPs subjected to ethanol washing displayed a polycrystalline arrangement, with an average crystal size measuring 21.3 nm. When the pH value was raised to 7.5 while still using ethanol as the washing solvent, the GNMIOPs exhibited a reduced average nanoparticle size, measuring 17.8 nm (as shown in Table 3), accompanied by an increase in the proportion of Fe_3_O_4_. Similarly, when the pH was further increased to 12.5, with ethanol as the washing solvent, the particles continued to decrease in size to 6.6 nm for GNMIOPs (Table 3), while maintaining an increased proportion of Fe_3_O_4_ in the samples (Table 2). Furthermore, when GNMIOPs, initially prepared at the pH of 7.5, underwent washing with water as the solvent, the measured average size was 23.5 nm (Table 3), indicating an increase in size with a decreased proportion of Fe_3_O_4_. The variation in particle size observed after washing with different solvents, such as ethanol or water, is primarily due to changes in surface modification and dispersion characteristics when the particles come into contact with these solvents. Additionally, it is worth emphasizing that the washing process can alter various properties, including crystal systems, oxide phase, crystallinity, and, as a result, the particle’s single crystal core. The size distribution of GNMIOPs was confirmed with SEM and TEM analyses.

### 3.2. SEM

Figure 3 illustrates the SEM images of the GNMIOPs prepared at different pH values (1.2, 7.5, and 12.5) and washed with ethanol or H_2_O (pH = 7.5). 

The GNMIOPs that were washed with ethanol displayed a variety of nanostructures, including cubic, face-centered rhombohedral, and hexagonal shapes. They also exhibited a more homogeneous size distribution with only slight agglomeration. The GNMIOPs prepared at pH = 7.5 (washed with H_2_O) exhibited spherical and flower-like morphologies with agglomerates/aggregates. The differences in agglomeration/aggregation observed in the various GNMIOPs can be attributed to the interaction between phenolic compounds and the surface of Fe_x_O_y_-NPs through H bonding, involving bioactive molecules [33], as well as interactions among the different crystalline systems present in the sample, as demonstrated in XRD.

The thresholding of SEM images revealed that the average diameter of the GNMIOPs ranged from 8.3 to 20.8 nm, which confirmed the trend observed in the XRD analysis (Table 3). These findings were further supported by the TEM results. 

### 3.3. TEM

Figure 4 shows TEM images of the GNMIOPs prepared at different pH values (1.2, 7.5, and 12.5) and washed with ethanol or H_2_O (pH = 7.5) with different morphologies. Their size distributions are also shown in Figure 4. 

The TEM analysis confirmed that the choice of solvent can significantly affect the morphology and size distribution of the nanoparticles. When the GNMIOPs were washed with ethanol, weak aggregation and cubic nanoparticles were observed, while quasi-spherical and cubic nanoparticles with a hexagonal axis were observed when washed with water. The various morphologies seen in SEM images are reflected in the XRD patterns, indicating structural disorder due to different morphologies or crystallographic orientations of Fe_x_O_y_-NPs.

The diameter of GNMIOPs was observed to range from 10.6 ± 0.2 to 22.2 ± 0.4 nm, which strongly agrees with the trends observed in the previous XRD and SEM analyses (Table 3). 

### 3.4. FTIR

The FTIR spectra of the different GNMIOPs are presented in Figure 5. Observations were conducted spanning the 800–400 cm^−1^ range, where magnetic Fe-O bands can be identified both as γ–Fe_2_O_3_/α–Fe_2_O_3_ and magnetite Fe_3_O_4_.

The FTIR spectra also confirmed the existence of various functional groups on the surface of the synthesized GNMIOPs, and the observed peaks were identified as various functional groups based on the literature. The FTIR spectra of GNMIOPs confirm the presence of Fe–O bands in the 800–400 cm^−1^ range, which are consistent with both magnetite Fe_3_O_4_ and maghemite γ–Fe_2_O_3_/α–Fe_2_O_3_. In comparison with maghemite, the spectrum of magnetite (Fe_3_O_4_) has bands at 640–570 cm^−1^, followed by a shoulder around 699 cm^−1^ and another shoulder around 448 cm^−1^ because of the Fe–O bond in the octahedral and tetrahedral positions, respectively [34]. The bands observed at 587–573 cm^−1^ indicate the reduction of α–Fe_2_O_3_ to Fe_3_O_4_ [35]. The Fe_3_O_4_ FTIR spectra show new absorption bands at 1627 and 1390 cm^−1^, as well as peaks at 1285 and 1085 cm^−1^ [36]. Nevertheless, the peculiar bonds observed at 740–640 cm^−1^ may have been due to the maghemite (γ–Fe_2_O_3_), resulting from the magnetite oxidation during synthesis [37]. The bands observed around 562, 540, and 462 cm^−1^ indicate the stretching vibration mode of Fe–O of the hematite phase (α–Fe_2_O_3_) [38]. The 1136 cm^−1^ signal indicates α–Fe_2_O_3_ phase, linked to Fe-O vibrations in the crystal lattice [39]. It is worth mentioning that despite the low pH, the reduction of Fe^3+^ (Fe_2_O_3_) to Fe^2+^ (Fe_3_O_4_) was mainly caused by the action of the polyphenolic compounds found in the extracts for the GNMIOPs [40].

The bonds at 3470–3422 cm^−1^ correspond to –OH vibrational stretching, found in polyphenol groups of the extract and NaOH–OH [41]. Two sharp peaks between 2912 and 2840 cm^−1^ indicate hydrocarbon extension. At 1631 cm^−1^, the band relates to aromatic ring deformation or C=C vibration in alkane groups. At 1734 cm^−1^, the band is assigned to the C=O bonds of aldehydes, ketones, and ester. At 1380 cm^−1^, the band is associated with ester groups [42]. The C–O and C–O–C asymmetric stretching vibrations, which are characteristic of polyphenol compounds, are observed between 1200 and 1247 cm^−1^, and between 1039 and 1070 cm^−1^, respectively [40]. Finally, the bonds in the range of 1160–1105 cm^−1^ are assigned to C–O–H in phenolic compounds [40]. The reduction of FeCl_3_·6H_2_O with the oxygen atoms of phenolic groups (–OH) was observed after dividing the 1642 cm^−1^ band into three different peaks (1623, 1633, and 1653 cm^−1^).

Differences in the peak intensities and positions in the FTIR spectra of the synthesized GNMIOPs were attributed to the effect of different pH levels and washing processes, which influenced the bonding characteristics of the MIONPs. For instance, the Fe–O band location was consistent with XRD results, and the reduction of Fe^3+^ to Fe^2+^ was mainly caused by the phenolic compounds found in the GNMIOPs’ extracts. The bands observed at 587–573 cm^−1^ indicate the reduction of α–Fe_2_O_3_ to Fe_3_O_4_ [35]. Nevertheless, the peculiar bonds observed at 740–640 cm^−1^ may have been due to the maghemite (γ–Fe_2_O_3_), resulting from the magnetite oxidation during synthesis [34].

### 3.5. Magnetic Properties

Figure 6 illustrates the hysteresis curves of GNMIOPs, displaying magnetization (*M*) as a function of magnetic field (*H*) at various pH levels (1.2, 7.5, and 12.5) and after the samples underwent washing with either ethanol or H_2_O (at pH = 7.5). 

The results obtained for the magnetic properties of GNMIOPs using different methods and pH values are further summarized in Table 4. 

The saturation magnetization (*M_s_*) values of GNMIOPs exhibited significant differences (*p* < 0.05) across all pH levels and washing solvents. The highest *M_s_* value was achieved at pH 7.5 when the sample was washed with ethanol.

The results of the magnetic properties of the GNMIOPs showed significant differences (*p* < 0.05) in their coercivity (*H_c_*) values across all pH values and washing solvents. The GNMIOPs prepared at pH 1.2 (washed with ethanol) exhibited the highest Hc value, followed by those prepared at pH 7.5 and 12.5 and washed with ethanol. On the other hand, the GNMIOPs prepared at pH 7.5 and washed with water showed the lowest *H_c_* value (Table 4). Furthermore, the remanence (*M_r_*) values of GNMIOPs at different pH values and washing solvents differed significantly for all conditions. The highest *M_r_* value for GNMIOPs was obtained at pH 1.2 (23.4 emu/g) using ethanol. Additionally, the *M_r_/M_s_* ratio values for GNMIOPs were found to be significantly different among all pH values and washing solvents, ranging from 0.21 to 0.41. 

These findings suggest that the magnetic behavior of the nanoparticles can be modulated by adjusting the pH and the washing process. 

The magnetic properties of the GNMIOPs were analyzed based on XRD, SEM, TEM, and FTIR results. At pH 1.2, when the GNMIOPs were washed with ethanol, they displayed a cubic hematite structure, with Fe_2_O_3_ constituting 78.2% of the nanoparticles. The presence of this cubic structure, along with the presence of Fe_3_O_4_–NPs, resulted in a saturation magnetization (*M_s_*) value of 57.5 emu/g. This suggests a preference for magnetic behavior in the cubic structure of the nanoparticles. With an increase in pH, while still washing with ethanol, iron hydroxide Fe(OH)_2_ precipitated, leading to the formation of magnetite and maghemite mixed phases (with Fe_3_O_4_ constituting 84.4%) at pH 7.5. This resulted in an increase in *M_s_* to 64.9 emu/g. A further increase in pH to 12.5 resulted in the formation of smaller, nearly pure magnetite nanoparticles with an *M_s_* value of 62.2 emu/g. The magnetic behavior was influenced by the Fe_3_O_4_:Fe_2_O_3_ ratio and the presence of cubic hematite and magnetite phases. However, the decrease in *M_s_* at pH 12.5 can be attributed to the presence of a trigonal with a hexagonal axis crystal system (1.1%), which was observed in the GNMIOPs prepared at pH 1.2 (86%) but not in those prepared at pH 7.5. This crystal system may promote interactions rather than exhibit magnetic behavior. The washing process affected the nanoparticle properties, with minimal or no aggregation being observed in GNMIOPs washed with ethanol at pH 7.5, while flower-like agglomeration and quasi-spherical structures were observed in GNMIOPs washed with water at pH 7.5. The decrease in saturation magnetization when the samples were washed with water could be attributed to nanoparticle aggregation and interactions. The solvent used for washing, such as ethanol or water, played a significant role in the aggregation behavior, with ethanol promoting better drying properties and reduced agglomeration [43]. The viscosity and surface tension of the solvents also influenced the nanoparticle formation process [44]. The remanence-to-saturation magnetization ratio was not applicable to fine particles due to interactions and surface effects [45,46]. Designing nanoparticles with desired magnetic properties requires considering various factors and their impact on nanoparticle characteristics and behavior. 

When comparing our findings and their significance to the different methods reported in the literature (as displayed in Table 5), it is important to consider various factors. 

In addition to the factors discussed above, it is noteworthy to mention that the reduced saturation magnetization (*M_s_*) of iron oxide nanoparticles compared with bulk Fe_3_O_4_ (93 emu/g) can be attributed to several additional factors [49]. Firstly, the presence of a non-magnetic layer on the surface of the nanoparticles, commonly referred to as a “dead magnetic layer”, contributes to the lower *M_s_*. Secondly, the distribution of cations within the nanoparticles and the surface spin disorder also play a significant role in reducing *M_s_*. These factors have been proposed as the main reasons behind the lower *M_s_* observed in nanoparticles compared with their bulk state. Hence, it is essential to take into account these various factors when interpreting and comparing our findings with the existing literature.

### 3.6. Nanoparticle Stability

The zeta potential measurements of iron oxide nanoparticles in distilled water are shown in Figure 7. 

Table 4 summarizes the zeta potential measurements of GNMIOPs at different pH values and washing solvents. All samples’ zeta potential values were negative, indicating that the particles were stable under the tested conditions. The zeta potential values for GNMIOPs were significantly different (*p* < 0.05) among all pH values and washing solvents. The GNMIOPs displayed zeta potential values between −41.6 and −31.8 mV.

The stability of irregularly shaped nanoparticles, as affected by their crystallinity and oxide phase, was found to be slightly influenced by changes in these parameters, with no clear relationship being observed. For instance, the GNMIOPs prepared at pH 12.5 and washed with ethanol, which exhibited the highest crystallinity (99%), displayed an intermediate zeta potential value of −35.1 mV. This value falls within the range associated with good stability [52]. 

However, these measurements were strongly correlated with the magnetic properties of the GNMIOPs. For instance, the GNMIOPs prepared at pH 7.5 and washed with ethanol, which had higher magnetization saturation (64.9 emu/g, Table 4), tended to exhibit lower stability (−31.8 mV; Table 4). On the other hand, the GNMIOPs prepared at pH 7.5 and washed with H_2_O, with lower *M_s_* (49.3 emu/g), showed the highest zeta potential (−41.6 mV). Similar findings have been reported in other studies involving similar particles, indicating that magnetic interactions among the nanoparticles can result in aggregation and clustering in the solution [52]. 

### 3.7. Antioxidant Activity

DPPH was used to measure the antioxidant activity of GNMIOPs. As shown in Figure 8, the prepared nanoparticles demonstrate antioxidant activity in function of pH in conjunction with their preparation method. 

The half-maximal inhibitory concentration (*IC_50_*) of GNMIOPs was measured against the DPPH free radical to evaluate their antioxidant capacity, and the values are summarized in Table 3. The *IC_50_* values of GNMIOPs exhibited noteworthy variations, being 5.6, 0.9, and 0.7 mg/mL at pH 1.2, 7.5, and 12.5, respectively, after ethanol washing, and 2.1 mg/mL at pH 7.5 following H_2_O washing. Importantly, the *IC_50_* values of all GNMIOPs demonstrated substantial variations, depending on the pH level and the type of washing solvent applied. The highest antioxidant activity was achieved by GNMIOPs prepared at pH 12.5 and 7.5 (Table 3). This may have been due to the simultaneous activity of polyphenols remaining as antioxidant agents and GNMIOPs as catalysts, a high proportion of Fe_3_O_4_, smaller size, higher crystallinity, and higher magnetic properties (*M_s_*). Similar trends have been found in previous investigations [53,54]. Nevertheless, when comparing the antioxidant activity of chemical iron oxide nanoparticles under the same conditions, we found that higher antioxidant effects were pronounced when using a green extract. For example, the sample prepared without using a polyphenol extract at pH 7.5 and washed with ethanol exhibited an *IC_50_* value of about 4.8 mg/mL [55], which is five times less effective than those prepared under the same conditions but using a polyphenol extract (0.9 mg/mL; Table 3).

The efficient ecological biosynthesis of GNMIOPs has important implications for their use as magnetically directed natural antioxidant additives in the pharmaceutical, biomedical, and food industries. The antioxidant property benefits many other properties, such as antibacterial, anti-inflammatory, anticancer, anti-obesity, and anti-viral properties [54], due to the capacity to produce reactive oxygen species (ROS), which can interact with functional proteins in a variety of ways, including specific and non-specific binding [56]. 

Green nanomagnetic iron oxide particles (GNMIOPs) function as antioxidants through multiple mechanisms (Figure 9).

Nanomagnetic iron oxide particles prepared using green methods often contain polyphenolic compounds on their surface, contributing to their antioxidant behavior. These nanoparticles exhibit antioxidant activity through various mechanisms, including hydrogen atom transfer (HAT) and single-electron transfer (SET). In HAT, the antioxidant reacts with free radicals by donating a hydrogen atom, forming a stabilized neutral species, while the antioxidant itself becomes an antioxidant-free radical [57] (Figure 9a). SET involves the formation of an energetically stable species, DPPH^−^, with an even number of electrons, while the cation radical of the antioxidant (R-OH^+•^) exhibits stability. The odd number of electrons formed during the reaction is distributed over the entire molecule, utilizing the aromatic ring structure and leading to radical stabilization [58] (Figure 9b). The antioxidant function of phenolic compounds depends on their structure, particularly the benzene ring, and the number and position of hydroxyl groups. The benzene ring plays a crucial role in stabilizing antioxidant molecules during their interaction with free radicals. For example, gallic acid, a phenolic acid with hydroxyl and carboxylic acid groups, exhibits antioxidant function by forming a stabilized gallic acid-free radical [59]. When considering the reduction of DPPH by iron oxide antioxidants such as Fe_2_O_3_ and Fe_3_O_4_, which exist as solid materials, the transfer of electrons from the antioxidant surface to the DPPH radical occurs, neutralizing it. The redox properties of iron ions present in the oxide structure facilitate this process. In the case of Fe_2_O_3_, Fe^3+^ ions in the 3+-oxidation state donate/accept electrons to/from DPPH, converting it to a stable molecule and causing a color change (Figure 9c). Fe_3_O_4_, a mixed-valence oxide, allows for direct electron donation from Fe^2+^ ions and regeneration of Fe^2+^ through redox cycling by Fe^3+^ ions, enabling the continuous reduction of DPPH (Figure 9d) [60,61]. The specific details of these mechanisms depend on factors such as surface area, crystal structure, particle size, magnetic behavior, DPPH concentration, and experimental conditions. Detailed chemical mechanisms illustrating the reduction of DPPH by iron oxide antioxidants are provided in Figure 9c,d, indicating the oxidation of iron oxide antioxidants, the generation of Fe^3+^ ions, electron donation to the DPPH radical, and formation of stable DPPH-H. These mechanisms may involve multiple steps and intermediates. The nanomagnetic iron oxide particles also demonstrated reusability, maintaining their antioxidant capacity over multiple cycles, indicating their potential shelf life.

### 3.8. Characterization of PCL/GNMIOP Membranes

#### 3.8.1. SEM Visualization

Figure 10 illustrates the morphologies of electrospun nanofiber membranes derived from PCL and PCL/GNMIOPs that were prepared under pH 7.5 conditions and subjected to either ethanol or water washing. 

The PCL membranes without GNMIOPs exhibited uniform and smooth nanofibers. However, the addition of GNMIOPs introduced irregularities and roughness to the nanofiber surface. This indicates the influence of GNMIOPs’ properties, such as dispersion, granulation, and agglomeration, on the fiber surfaces during electrospinning. The presence of GNMIOPs resulted in broader fiber zones, particularly with GNMIOPs prepared at pH 7.5 and washed with water, as well as a wider dispersion of fiber diameters with larger aggregation knots. In contrast, those washed with ethanol showed fewer aggregation knots. Several factors can be attributed to this behavior, including particle size, crystallinity, and hydrophobicity. Previous studies have also reported the formation of larger nanoparticle aggregates in systems with higher nanoparticle hydrophobicity [62]. Figure 11 shows the distribution of fiber diameters for both PCL membranes without and with GNMIOPs incorporated.

Additionally, introducing GNMIOPs prepared at pH 7.5 and subjected to either ethanol or water washing led to variations in the average nanofiber diameter, highlighting the unique characteristics of these nanoparticles (as shown in Table 6). 

#### 3.8.2. TEM

Figure 12 illustrates the TEM image morphologies of electrospun nanofiber membranes derived from PCL and PCL/GNMIOPs, which were prepared under pH 7.5 conditions and subjected to either ethanol or water washing.

It is evident that GNMIOPs that were prepared at pH 7.5 and subjected to ethanol washing demonstrated superior dispersion and a more even distribution compared with those washed with water. They exhibited reduced agglomeration. Conversely, GNMIOPs prepared at pH 7.5 and washed with water showed higher aggregation, which could be attributed to the increased hydrophobic nature acquired during the washing process. Another possible explanation is that aggregation is a result of the interactions between the charged phenolic compounds on the GNMIOP surfaces and the PCL chains within the fibers, leading to changes in their structures [62]. This phenomenon holds significance regarding surface roughness, as both variants of GNMIOPs play a role in modifying the surface of the fibers. Further roughness measurements were conducted and are discussed in the following section.

#### 3.8.3. Surface Roughness

Two-dimensional surface plots of electrospun nanofiber membranes derived from PCL and PCL/GNMIOPs that were prepared under pH 7.5 conditions and subjected to either ethanol or water washing are depicted in Figure 13. 

The roughness results, represented by *Ra* and *RMS* values, are summarized in Table 6. The inclusion of GNMIOPs, whether washed with ethanol or water, substantially increased both *Ra* and *RMS* values compared with the PCL membrane alone. The most significant roughness enhancement occurred in GNMIOPs subjected to water washing, primarily due to larger aggregates within the fibers resulting from the larger size of water-washed GNMIOPs. Additionally, interactions among different nanoparticle crystalline systems played a key role in forming these larger aggregates. These findings are consistent with the SEM and TEM analyses and are supported by previous studies [63,64]. The roughness of the membrane surfaces plays a critical role in the growth and adherence of different cell lines, making it an important parameter for biomaterial applications. 

#### 3.8.4. Fiber Magnetic Properties

Figure 14 shows the hysteresis curves of magnetization (M) against the magnetic field (H) of electrospun nanofiber membranes derived from PCL and PCL/GNMIOPs that were prepared under pH 7.5 conditions and subjected to either ethanol or water washing.

The investigation of magnetic properties in the electrospun nanofiber membranes revealed interesting findings. The sample with PCL alone did not exhibit any magnetic properties. However, magnetic behaviors became evident upon the incorporation of GNMIOPs into the PCL matrix. Specifically, the PCL/GNMIOPs prepared at pH 7.5 and washed with ethanol displayed saturation magnetization (*M_s_*) of 1.8 emu/g, remnant magnetization (*M_r_*) of 0.7 emu/g, and coercivity (*H_c_*) of 10.9 mT, indicating the presence of magnetic properties. In contrast, the PCL/GNMIOPs prepared at pH 7.5 and washed with water showed a significant reduction in magnetic properties. The Ms value decreased to 0.9 emu/g; the *M_r_* value decreased to 0.1 emu/g; and the *H_c_* value increased to 28.3 mT (Table 6). This suggests a decrease in the magnetic response of the sample. It is important to note that these observed magnetic behaviors were obtained upon the incorporation of 1% *w*/*w* GNMIOPs. By increasing the concentration and considering other factors such as composition and processing parameters, a further tailoring of the magnetic properties can be achieved.

These results demonstrate the influence of pH value and washing solvent on the magnetic properties of the nanofiber membranes. Moreover, it highlights the significance of the interactions between GNMIOPs and the surrounding PCL environment in determining the observed magnetic behavior.

#### 3.8.5. Free Radical Inhibition Percentage (FRIP, %)

The incorporation of GNMIOPs into the PCL matrix led to a significant increase in free radical inhibition percentage (FRIP) compared with pure PCL (Table 6). This enhancement can be attributed to various factors. Firstly, GNMIOPs, whether washed with ethanol or water, contain phytochemical compounds known for their antioxidant properties that contribute to the improvement in FRIP [65,66]. Additionally, factors such as the magnetic behavior and smaller size of the GNMIOPs, along with a higher proportion of magnetite, further contribute to the observed enhancement. The GNMIOPs exhibited smaller sizes, better magnetic behavior, and higher magnetite content, resulting in a significant increase in FRIP. These findings highlight the combined effect of phytochemical compounds, magnetic behavior, and size on the antioxidant activity of PCL/GNMIOP nanocomposites. The enhanced antioxidant capacity of GNMIOPs incorporated into the PCL matrix offers potential benefits for various other properties. These properties include antimicrobial, anticancer, and antibiotic activities, as evidenced by previous research findings [67]. The antioxidant activity of GNMIOPs can synergistically contribute to these additional beneficial properties. The ability to scavenge free radicals and neutralize oxidative stress can aid in combating microbial infections, inhibiting cancer cell growth, and supporting the effectiveness of antibiotic treatments. The multifaceted nature of GNMIOPs, with their antioxidant potential and demonstrated impact on other properties, presents promising opportunities for further exploration and applications in diverse fields.

## 4. Conclusions

Particle aggregation is promoted by low pH and water washing, which in turn has a significant impact on the magnetic properties of GNMIOPs. Both crystalline anisotropy and aggregation influence saturation magnetization, coercivity, and remanence. Further research is needed to explore the effects of alternating magnetic fields and temperature on GNMIOPs. By controlling pH values and solvents, aggregation and crystallinity can be manipulated to enhance magnetism. Green synthesis at higher pH values and ethanol washing yield magnetite with improved properties. This study emphasizes the significant influence of pH and the choice of washing solvent on magnetic iron oxide nanoparticles, offering valuable insights for a range of applications. Moreover, GNMIOPs influence nanofiber morphology, roughness, and magnetic behavior in electrospun membranes. Ethanol washing enhances dispersion and magnetic response, while water washing reduces magnetism. The exceptional magnetic and antioxidant properties of GNMIOPs and their environmental and health safety make them an excellent alternative to conventional synthesis. These green GNMIOPs are valuable as natural magnetic antioxidant agents with multiple applications, including biomedical, pharmaceutical, food, electronic, and chemical industries. They serve as biodrugs and carriers in complex systems, making a significant contribution to the field. Furthermore, when incorporated into PCL nanofibers, they exhibit magnetic properties, opening up additional possibilities for applications in areas such as tissue engineering, drug delivery systems, and magnetic sensing devices. In future studies, we will conduct a more comprehensive quantitative analysis of polyphenol content in nanoparticle-based green synthesis and evaluate its influence on our research outcomes.

## Figures and Tables

**Figure 1 polymers-15-03850-f001:**
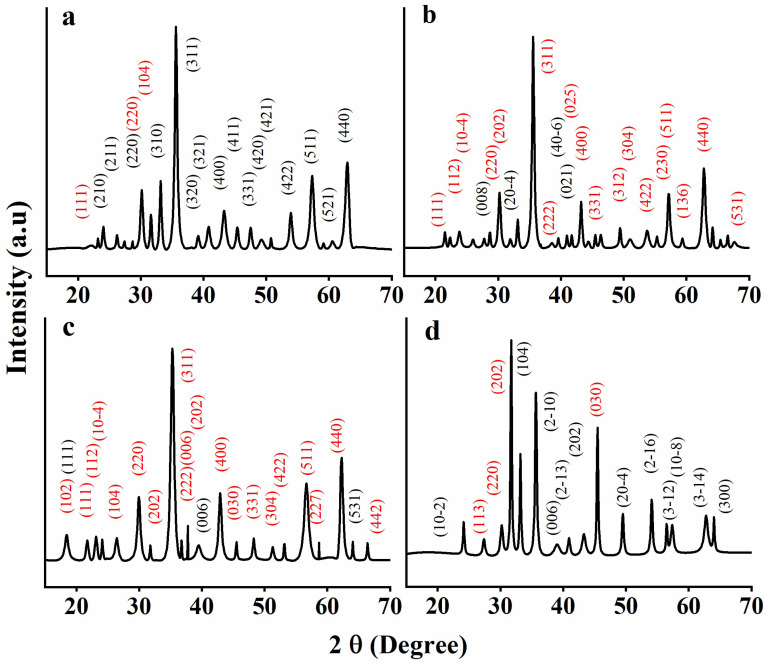
XRD spectra of GNMIOPs prepared at different pH levels (1.2, 7.5, and 12.5) and washed with ethanol ((**a**), (**b**), (**c**), respectively), and GNMIOPs prepared at pH = 7.5 and washed with H_2_O (**d**). Hematite is represented by black planes, while magnetite is represented by red planes (based on the JCPDS standard).

**Figure 2 polymers-15-03850-f002:**
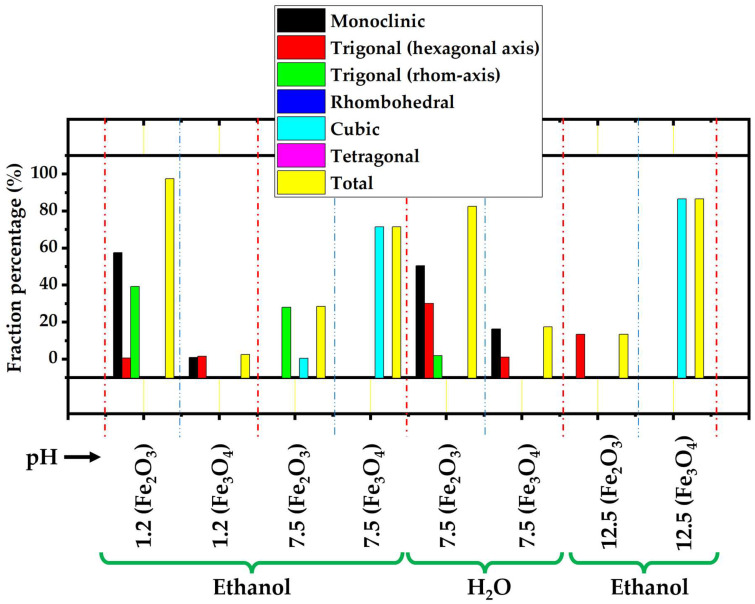
The different crystal systems attributed to Fe_2_O_3_ and Fe_3_O_4_ proportions of GNMIOPs prepared at different pH values (1.2, 7.5, and 12.5) and washed with ethanol or H_2_O (pH = 7.5). Commencing and concluding conditions are denoted by red dashed lines, while phase separation within a single condition is represented by blue dashed lines.

**Figure 3 polymers-15-03850-f003:**
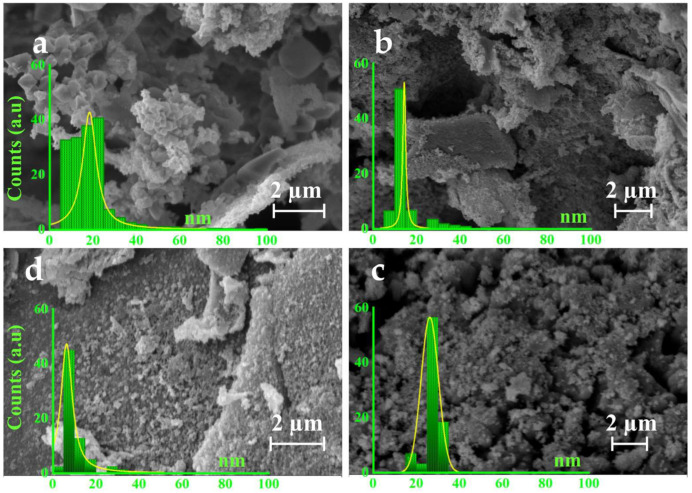
Scanning electron microscopy (SEM) images and diameter distributions (nm) of GNMIOPs prepared at different pH levels (1.2, 7.5, and 12.5) and washed with ethanol ((**a**), (**b**), and (**c**), respectively), and GNMIOPs prepared at pH = 7.5 and washed with H_2_O (**d**).

**Figure 4 polymers-15-03850-f004:**
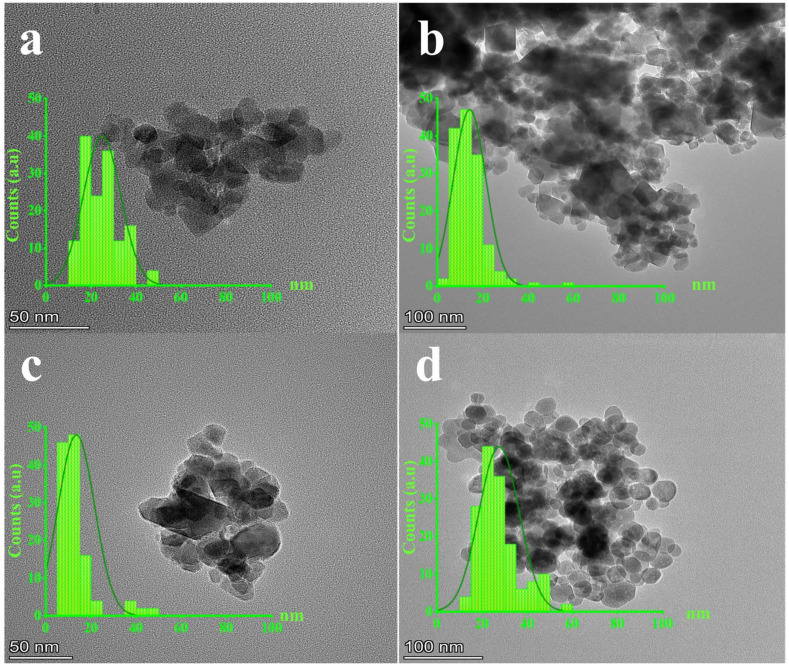
Transmission electron microscopy (TEM) images and diameter distributions (nm) of GNMIOPs prepared at different pH levels (1.2, 7.5, and 12.5) and washed with ethanol ((**a**), (**b**), and (**c**), respectively), and GNMIOPs prepared at pH = 7.5 and washed with H_2_O (**d**).

**Figure 5 polymers-15-03850-f005:**
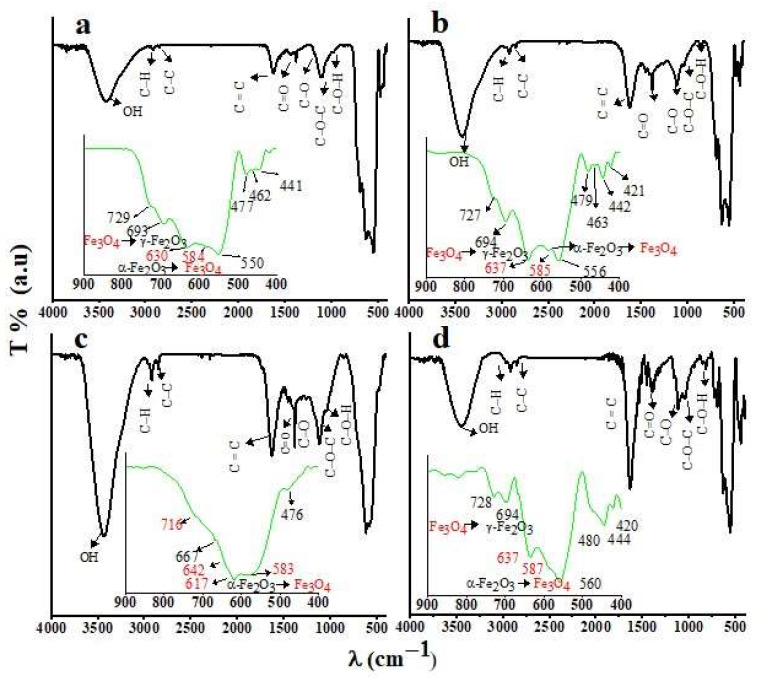
FTIR spectra of GNMIOPs prepared at different pH levels (1.2, 7.5, and 12.5) and washed with ethanol ((**a**), (**b**), (**c**), respectively), and GNMIOPs prepared at pH = 7.5 and washed with H_2_O (**d**). Green lines represent the FTIR spectra in the fingerprint region 800–400 cm^−1^, with red font indicating magnetite and black font indicating hematite phases.

**Figure 6 polymers-15-03850-f006:**
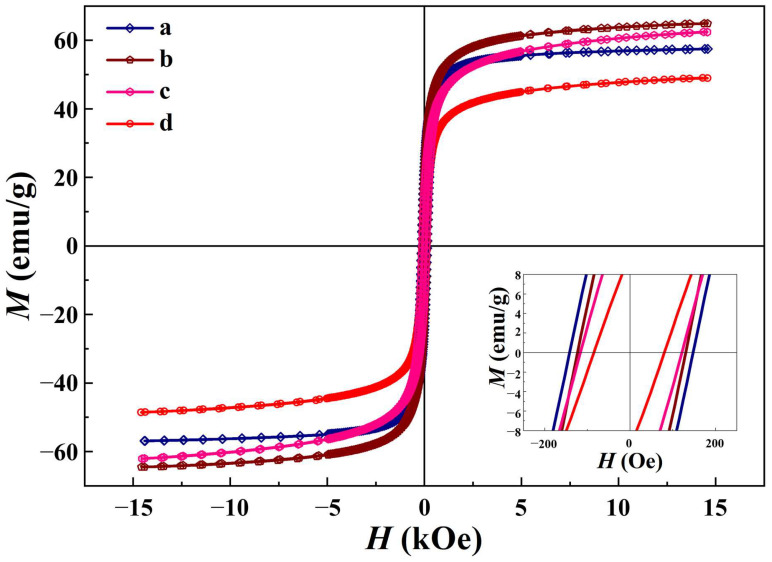
Magnetic hysteresis curves at room temperature of GNMIOPs prepared at different pH levels (1.2, 7.5, and 12.5) and washed with ethanol (a, b, c, respectively), and GNMIOPs prepared at pH = 7.5 and washed with H_2_O (d).

**Figure 7 polymers-15-03850-f007:**
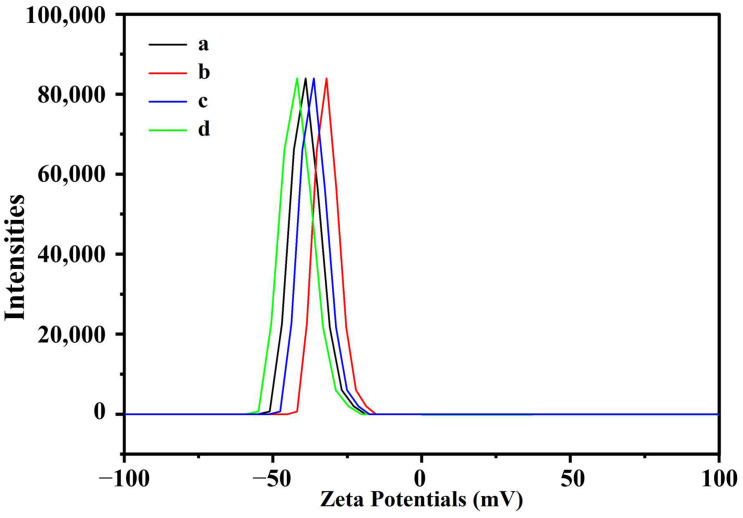
Zeta potential values of GNMIOPs at varied pH levels (1.2, 7.5, and 12.5) after ethanol washing (a, b, c, respectively) and GNMIOPs at pH = 7.5 following H_2_O washing (d).

**Figure 8 polymers-15-03850-f008:**
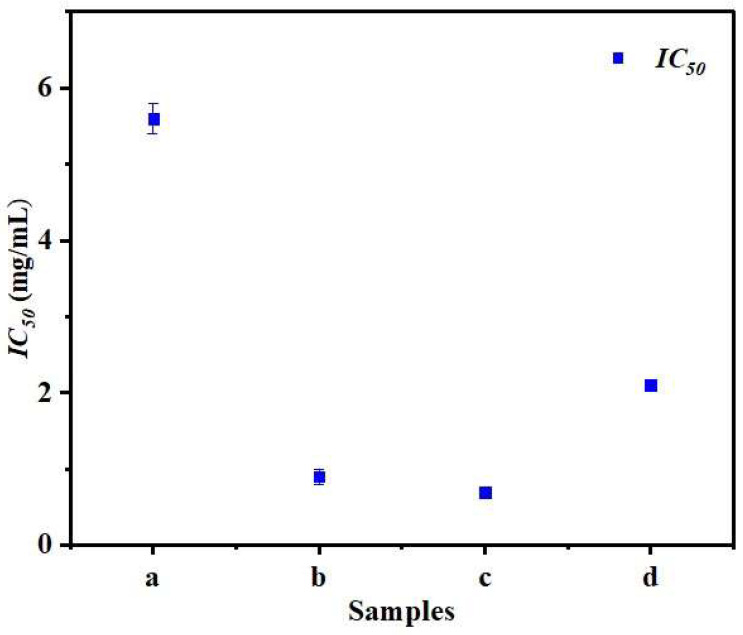
Antioxidant activity expressed as *IC_50_* (mg/mL) of GNMIOPs prepared at different pH levels (1.2, 7.5, and 12.5) and washed with ethanol (a, b, c, respectively), and GNMIOPs prepared at pH = 7.5 and washed with H_2_O (d).

**Figure 9 polymers-15-03850-f009:**
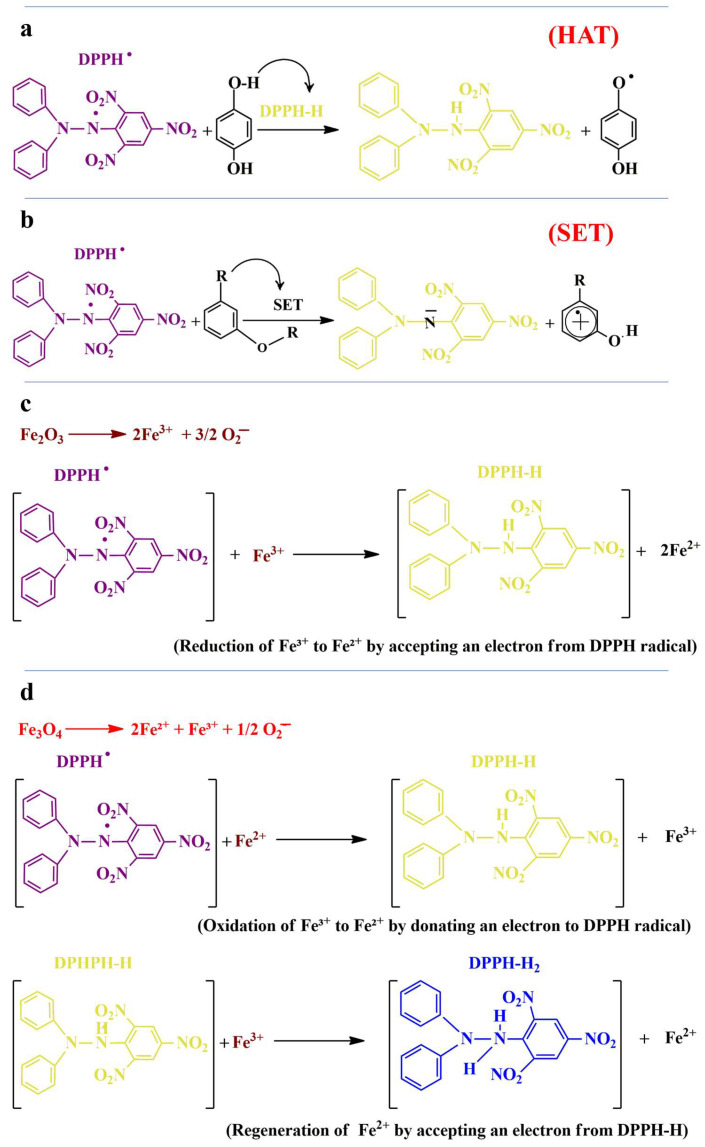
Antioxidant mechanisms of magnetic iron oxide nanoparticles (NMIOPs) in the presence of phenolic antioxidants. (**a**) Hydrogen atom transfer (HAT) mechanism. (**b**) Single-electron transfer (SET) mechanism. (**c**) SET mechanism of Fe_2_O_3_ antioxidants. (**d**) SET mechanism of Fe_3_O_4_ antioxidants.

**Figure 10 polymers-15-03850-f010:**
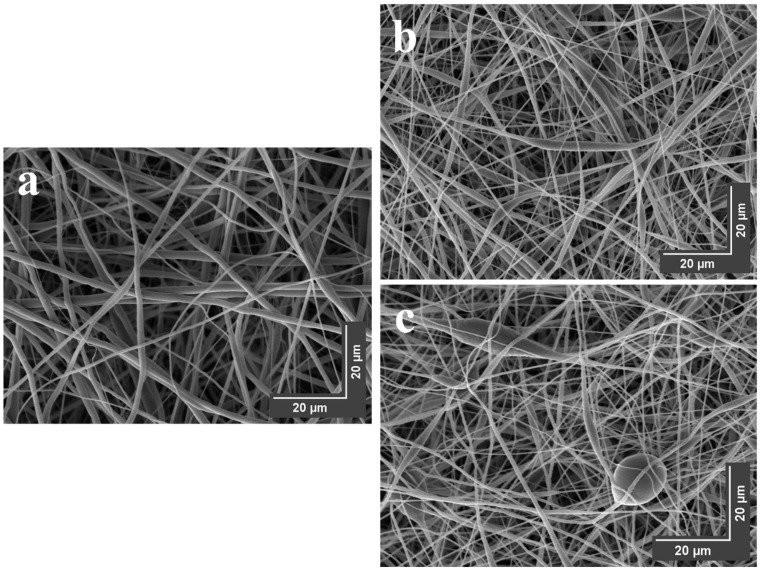
SEM images of electrospun nanofiber membranes derived from pure PCL (**a**) and PCL/GNMIOPs that were prepared under pH 7.5 conditions and subjected to either ethanol (**b**) or water washing (**c**).

**Figure 11 polymers-15-03850-f011:**
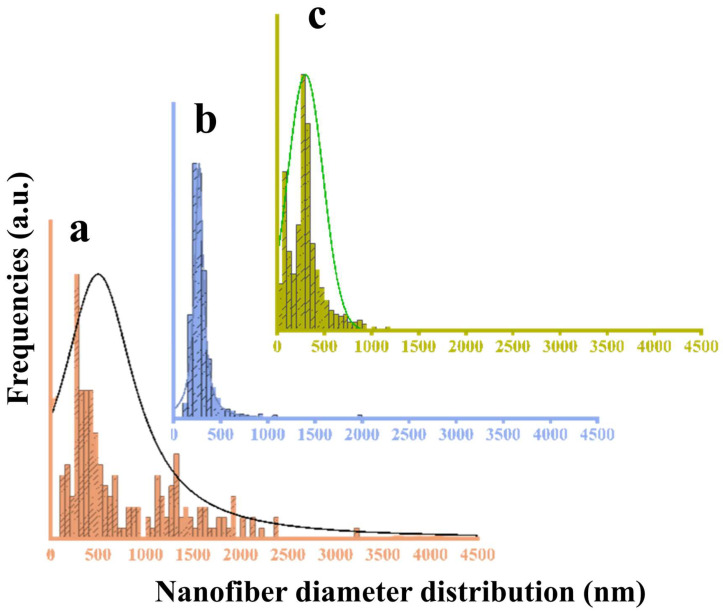
Diameter distributions of electrospun nanofiber membranes derived from pure PCL (**a**) and PCL/GNMIOPs that were prepared under pH 7.5 conditions and subjected to either ethanol (**b**) or water washing (**c**).

**Figure 12 polymers-15-03850-f012:**
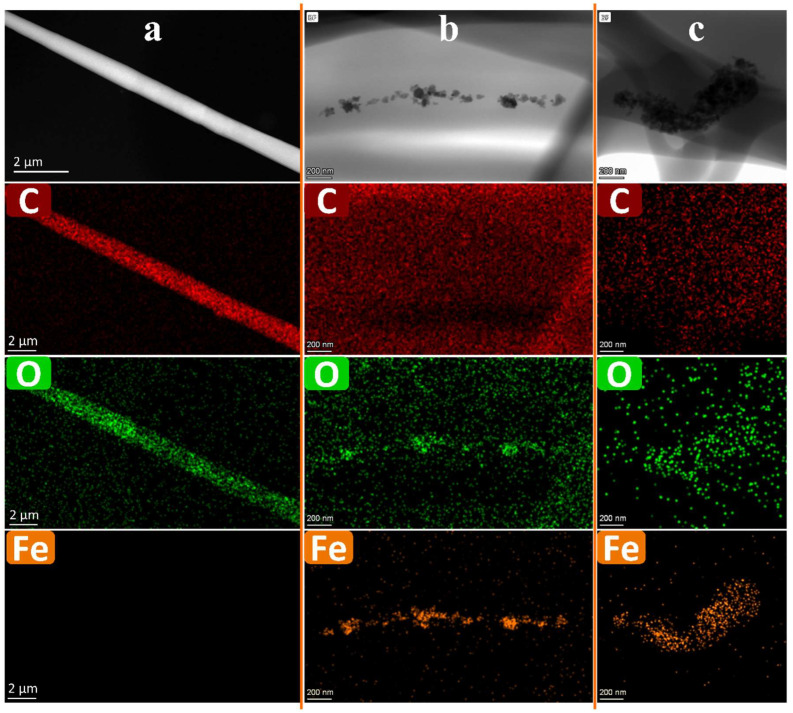
TEM images of electrospun nanofiber membranes derived from pure PCL (**a**) and PCL/GNMIOPs that were prepared under pH 7.5 conditions and subjected to either ethanol (**b**) or water washing (**c**).

**Figure 13 polymers-15-03850-f013:**
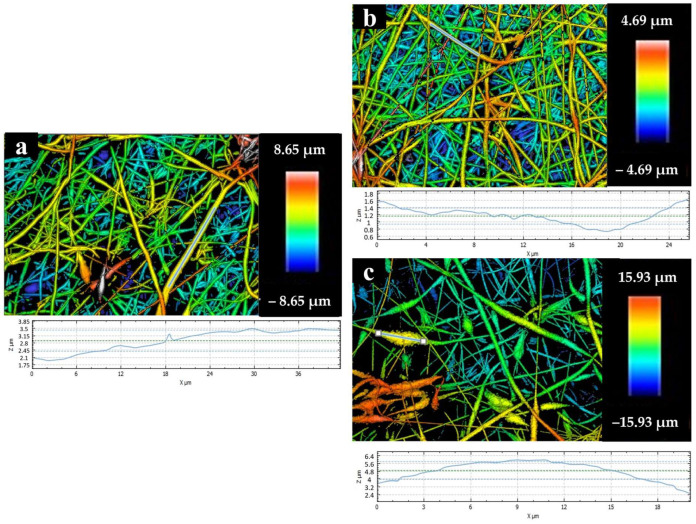
Two-dimensional surface plots of electrospun nanofiber membranes derived from pure PCL (**a**) and PCL/GNMIOPs that were prepared under pH 7.5 conditions and subjected to either ethanol (**b**) or water washing (**c**). Additionally, profiles of selected zones are included beneath each plot, providing information about specific regions of interest.

**Figure 14 polymers-15-03850-f014:**
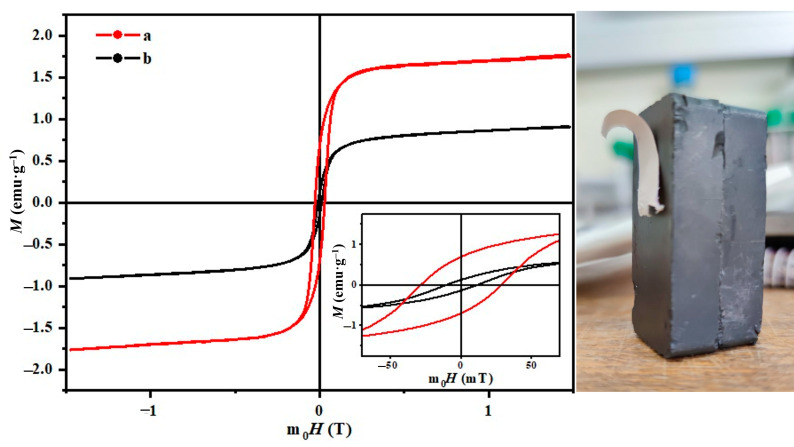
Magnetic hysteresis curves at room temperature of electrospun nanofiber membranes obtained from PCL/GNMIOPs prepared at pH 7.5 and washed with ethanol (a) and PCL/GNMIOPs prepared at pH = 7.5 and washed with H_2_O (b). Pure PCL did not exhibit any magnetic properties. The figure on the right visually illustrates the membrane’s response to the magnet (sample a).

**Table 1 polymers-15-03850-t001:** Results obtained for 2θ (°) and corresponding crystallographic reflection planes of GNMIOPs prepared at different pH values (1.2, 7.5, and 12.5) and washed with ethanol or H_2_O (pH = 7.5). Fe^2+^ indicates magnetite (Fe_3_O_4_), and Fe^3+^ indicates hematite (Fe_2_O_3_).

GNMIOPs							
1.2		7.5				12.5	
		Ethanol		H_2_O			
2θ (°)	Planes	2θ (°)	Planes	2θ (°)	Planes	2θ (°)	Planes
18.30	(111) Fe^2+^	21.55	(111) Fe^2+^	24.12	(10-2) Fe^3+^	18.13	(111) Fe^3+^
23.20	(210) Fe^3+^	23.83	(112) Fe^2+^	27.38	(113) Fe^2+^	18.31	(102) Fe^2+^
26.23	(211) Fe^3+^	25.99	(10-4) Fe^2+^	30.23	(220) Fe^2+^	21.76	(111) Fe^2+^
30.17	(220) Fe^3+^	28.71	(008) Fe^3+^	31.70	(202) Fe^2+^	23.15	(112) Fe^2+^
30.17	(220) Fe^2+^	30.23	(220) Fe^2+^	33.17	(104) Fe^3+^	24.12	(10-4) Fe^2+^
31.64	(−104) Fe^2+^	33.12	(202) Fe^2+^	35.68	(2-10) Fe^3+^	26.44	(104) Fe^2+^
33.17	(310) Fe^3+^	35.56	(20-4) Fe^3+^	39.02	(006) Fe^3+^	30.04	(220) Fe^2+^
35.62	(311) Fe^3+^	38.57	(311) Fe^2+^	40.94	(2-13) Fe^3+^	31.80	(202) Fe^2+^
39.17	(320) Fe^3+^	39.59	(222) Fe^2+^	43.29	(202) Fe^3+^	35.31	(311) Fe^2+^
40.82	(321) Fe^3+^	41.74	(40-6) Fe^3+^	45.47	(030) Fe^2+^	36.90	(222) Fe^2+^
43.29	(400) Fe^3+^	43.22	(021) Fe^3+^	49.51	(20-4) Fe^3+^	37.11	(202) Fe^2+^
45.40	(411) Fe^3+^	44.37	(400) Fe^2+^	54.09	(2-16) Fe^3+^	38.72	(116) Fe^2+^
47.52	(331) Fe^3+^	45.47	(030) Fe^2+^	56.48	(3-12) Fe^3+^	39.50	(006) Fe^3+^
49.26	(420) Fe^3+^	46.34	(331) Fe^2+^	57.40	(10-8) Fe^3+^	43.05	(400) Fe^2+^
50.79	(421) Fe^3+^	49.44	(312) Fe^2+^	62.77	(3-14) Fe^3+^	45.52	(030) Fe^2+^
53.91	(422) Fe^3+^	51.04	(304) Fe^2+^	64.01	(300) Fe^3+^	48.74	(311) Fe^2+^
57.34	(511) Fe^3+^	53.74	(422) Fe^2+^			51.30	(304) Fe^2+^
60.57	(521) Fe^3+^	55.32	(230) Fe^2+^			53.18	(422) Fe^2+^
		57.15	(511) Fe^2+^			56.66	(511) Fe^2+^
		59.36	(136) Fe^2+^			59.07	(227) Fe^2+^
		62.78	(440) Fe^2+^			62.29	(440) Fe^2+^
		64.19	(410) Fe^2+^			64.07	(531) Fe^3+^
		65.43	(412) Fe^2+^			66.41	(442) Fe^2+^
		66.58	(531) Fe^2+^				
		67.62	(332) Fe^2+^				

**Table 2 polymers-15-03850-t002:** Results obtained for different crystal systems attributed to Fe_2_O_3_ and Fe_3_O_4_ proportions of GNMIOPs prepared at different pH values (1.2, 7.5, and 12.5) and washed with ethanol or H_2_O (pH = 7.5). Fe^3+^ indicates hematite (Fe_2_O_3_), and Fe^2+^ indicates magnetite (Fe_3_O_4_).

Samples			GNMIOPs					
pH	1.2		7.5				12.5	
			Ethanol		H_2_O			
Crystal system/phase (%)	Fe^3+^	Fe^2+^	Fe^3+^	Fe^2+^	Fe^3+^	Fe^2+^	Fe^3+^	Fe^2+^
Monoclinic		6.0	15.6			19.0		4.3
Trigonal (hexagonal axis)		8.6			62.4	1		1.1
Trigonal (rhombohedral axis)								
Rhombohedral								
Cubic	78.2	7.2		84.4		17	3.4	91.2
Tetragonal						0.6		
Total	78.2	21.8	15.6	84.4	62.4	37.6	3.4	94.6

**Table 3 polymers-15-03850-t003:** Results obtained for different parameters of GNMIOPs prepared at different pH values (1.2, 7.5, and 12.5) and washed with ethanol or H_2_O (pH 7.5): crystallinity percentage, average size in nm (from XRD measurements, and SEM and TEM images), and antioxidant activity (*IC_50_* DPPH free radical in mg/mL).

Sample	GNMIOPs			
pH/Washing Solvent	1.2	7.5		12.5
		Ethanol	H_2_O	
Crystallinity%	89.9	95.2	91.1	99.0
D _XRD_ (nm)	21.3 ± 1.4 ^b^	17.8 ± 1.0 ^c^	23.5 ± 0.6 ^a^	6.6 ± 4.9 ^d^
D _SEM_ (nm)	18.6 ± 1.1 ^b^	14.2 ± 0.5 ^c^	20.8 ± 0.1 ^a^	8.3 ± 0.9 ^d^
D _TEM_ (nm)	20.7 ± 1.0 ^b^	12.4 ± 0.5 ^c^	22.2 ± 0.4 ^a^	10.6 ± 0.2 ^d^
*IC_50_* (mg/mL)	5.6 ± 0.2 ^a^	0.9 ± 0.1 ^c^	2.1 ± 0.0 ^b^	0.7 ± 0.0 ^d^

Different superscript letters (^a–d^) within a row indicate significant differences among mean observations (*p* < 0.05).

**Table 4 polymers-15-03850-t004:** Results of magnetic properties of GNMIOPs prepared at different pH levels (1.2, 7.5, and 12.5) and washed with ethanol or H_2_O (pH = 7.5): saturation magnetization (*M_s_*, emu/g), coercivity (*H_c_*, Oe), remanence (*M_r_*, emu/g), and zeta potential (mV).

Sample	GNMIOPs			
Parameter	1.2	7.5		12.5
		Ethanol	H_2_O	
*M_s_* (emu/g)	57.5 ^c^	64.9 ^a^	49.3 ^d^	62.4 ^b^
*H_c_* (Oe)	144.8 ^a^	126.4 ^b^	82.4 ^d^	120.4 ^c^
*M_r_* (emu/g)	23.4 ^a^	21.7 ^b^	10.2 ^d^	16.0 ^c^
*M_r_*/*M_s_*	0.41	0.33	0.21	0.26
Zeta potential (mV)	−38.9 ^b^	−31.8 ^d^	−41.6 ^a^	−35.1 ^c^

Distinct superscript symbols (^a–d^) within the same row signify statistically significant variations among mean observations (*p* < 0.05).

**Table 5 polymers-15-03850-t005:** Comparison of magnetic properties of GNMIOPs obtained in this study with other methods reported in the literature.

Synthesis Method	*M_s_* (emu/g)	*H_c_* (Oe)	*M_r_* (emu/g)	Size (nm)	Phase	Reference
GNMIOPs (pH 1.2)	57.5	144.8	23.4	21.3	Fe_2_O_3_ (78.2%)	This study
GNMIOPs (pH 7.5)	64.9	126.4	21.7	17.8	Fe_3_O_4_ (84.4%)	This study
GNMIOPs (pH 12.5)	62.4	120.4	16.0	6.6	Fe_3_O_4_ (94.6%)	This study
GNMIOPs (pH 7.5, H_2_O)	49.3	82.4	10.2	23.5	Fe_3_O_4_ (62.4%)	This study
Hydrothermal	1.9	415.0	0.7	52.1	Fe_2_O_3_	[47]
Hydrothermal	58.6	330.4	16.7	13.2	γ–Fe_2_O_3_	[47]
Hydrothermal	40.3	381.6	13.1	14.3	Fe_3_O_4_	[47]
Green	60	-	110	13.0	FeO	[48]
Green	50	-	-	-	Fe_3_O_4_	[49]
Co-precipitation	2	-	155	23.0	FeO	[48]
Modified sol–gel	47	-	0.7	8.0	Fe_3_O_4_	[50]
Hydrolysis	55.4	-	9.8	9.0	Fe_3_O_4_	[51]

**Table 6 polymers-15-03850-t006:** Average nanofiber diameter (D_avg_, nm), *M_s_* (emu/g), *M_r_* (emu/g), *H_c_* (mT), quadratic root mean square average roughness (*RMS*, nm), roughness (*Ra*, nm), and free radical inhibition percentage (FRIP, %) for electrospun nanofiber membranes derived from pure PCL (a) and PCL/GNMIOPs that were prepared under pH 7.5 conditions and subjected to either ethanol (b) or water washing (c).

Sample	D_avg_ (nm)	*M_s_* (emu/g)	*M_r_* (emu/g)	*H_c_* (mT)	*RMS* (nm)	*Ra* (nm)	FRIP (%)
a	253 ± 109	-	-	-	21 ± 13	18 ± 10	21
b	288 ± 124	1.8	0.7	10.9	72 ± 10	56 ± 8	58
c	365 ± 178	0.9	0.1	28.3	85 ± 26	73 ± 22	52

## Data Availability

All data generated or analyzed during this study are included in this published article. Any further specific data analysis can be obtained by making a reasonable request to the corresponding author.

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
