# Peer review of "Sustainable Nanomagnetism: Investigating the Influence of Green Synthesis and pH on Iron Oxide Nanoparticles for Enhanced Biomedical Applications"

_polymers, 2023, doi:10.3390/polym15183850_

Round 1
Reviewer 1 Report
The authors present synthesis and properties of the iron oxide nanoparticles, prepared in the presence of polyphenolic compounds. Despite large quantity of similar studies, the authors made some new findings and the manuscript can be published, but in my opinion some serious issues should be addressed. Please find below my comments.
1. I'm surprised that iron oxide particles formed at pH = 1.2. This is strongly acidic medium and iron oxide should dissolve. Please check. Was it initial pH of the reaction mixture or final pH, did precipitation really occur at pH = 1.2?
2. Why does the size of the particles depend on washing solvent (please compare particles prepared at pH = 7.5 and washed with ethanol or water)? The particles already formed before they were washed, didn't they? This questions is especially interesting for the size determined by XRD – this is a size of the single crystal core, which can't change upon washing.
3. The particles studied by the authors contain some quantity of polyphenolic compounds, as evidenced by IR spectroscopy. What is real content of such polyphenols? Was it taken into account for analysis of magnetization? The difference between Ms (in emu/g) can be caused by the impurities of polyphenols, which "dilute" the magnetic phase.
4. To continue comment 3, what is the origin of the antioxidant properties of the samples? Can these properties be caused by the residues of polyphenol? The authors write about "simultaneous activity of polyphenols remaining as antioxidant agents and GNMIOPs as catalysts, high proportion of Fe3O4.." – in my opinion, the presence of polyphenols can be the sole reason for the antioxidant activity, and the difference between sample can be caused by different content of the polyphenols. Control experiment should be carried out – the antioxidant properties of the samples of iron oxides, prepared without polyphenols, should be measured to be sure that the authors study the effect of iron oxide, not the polyphenolic component.
5. The authors introduce the terms "crystallinity degree" and "crystallinity percentage". Do these terms have the same meaning or not? Anyhow, they should be defined, the method of calculation should be disclosed.
6. Crystallinity of the samples is provided with 0.1 % accuracy. Such high accuracy should be justified.
7. For comparison the magnetic properties of the authors' samples and the samples, reported in literature, particles size should be considered (Table 5). Some difference can be explained by different size of the particles.
8. The authors claim in the title, that their materials can be suitable for biomedical applications. This potential use should be discussed in more details.
9. Table 1 can be moved to Supplementary materials.
In addition, the style of manuscript writing requires serious improvement, because the manuscript contains a lot of misleading and incorrect statements. Some examples are provided below, but please pay attention that it is not a complete list.
Title: pH is a part of synthetic method, and I'm not sure that there are any reasons to separate pH and other components of synthetic methods. More important, the authors studied the influence of pH and washing procedure on the properties of samples, nothing else. What is a meaning of "synthetic method" in the title?
Abstract: the authors use the phrase "green nanomagnetic iron oxide particles (GNMIOPs)". Later in the text they use abbreviation "nanomagnetic iron oxide particles (NMIOPs)" along with GNMIOPs. I'm not sure that it is really important to use these two abbreviations. Moreover, in my opinion it would be more correct to call the particles as "magnetic iron oxide nanoparticles" instead of "nanomagnetic iron oxide particles", because the term "nanoparticle" is strictly defined, while "nanomagnetic" does not have generally accepted definition.
"The stability of GNMIOPs 27 ranges from -31.8 to -41.6 mV" – in fact, the stability is not measured in mV.
Introduction, paragraph[h 2: "Metal-polymer nanocomposites, which combine the unique optical, catalytic, electrical, magnetic, and functional characteristics of metal nanoparticles and polymer nanofibers, have gained significant attention." In fact, all listed here characteristics are functional characteristics, and the authors write something like "combine unique functional characteristics and functional characteristics"
Introduction, paragraph 3: "NMIOPs possess magnetic, specific, unique, and biocompatible features." In fact, all of these feature can be unique – it is not clear, which "unique feature" the authors speak about.
Later, "NMIOPs are safe, non-cytotoxic, and potential oral therapy for iron deficiency" – something is missing in this sentence.
I selected answer " Extensive editing of English language required" - I think that in many cases the problem is not English itself, but the style of the manuscript writing. A presented examples in my review.
Author Response
Dear Reviewer,
We want to express our sincere gratitude for dedicating your time and expertise to review our manuscript. Your thoughtful insights and feedback have been invaluable in improving the quality of our work.
Please find attached our response to your constructive comments. We have carefully addressed each of your suggestions, and your input has undoubtedly strengthened our manuscript.
Once again, thank you for your commitment to the peer-review process and for helping us enhance the overall quality of our research. Your contributions are greatly appreciated.
Warm regards,
Johar Amin Ahmed Abdullah

Reviewer 2 Report
This is an interesting study and may be accepted for the publication after minor revision.
1.As the title shows that "Sustainable Nanomagnetism: Investigating the Influence of 2 Synthesis Method and pH on Iron Oxide Nanoparticles for En- 3 hanced Biomedical Applications" but the experimental evidence is only shown by anti-oxidant assay. They need to conduct at least an in vitro assay or MTT assay to show the impact of prepared nanoparticles on animal or human cells.
2. They need to describe various biomedical applications where these prepared nanoparticles can be used
3. They need to explain how nanomagnetism impacted on anti-oxidant activity with suitable citations
Minor spell checking
Author Response

(The authors gave the same response as above.)

Round 2
Reviewer 1 Report
The authors took into account my comments and suggestions and in my opinion thei did their best for improving the manuscript.
Some minor issues should be corrected; however I think it will be fixed upon editing in Editorial office